# The Impact of Westernization on the Insulin/IGF-I Signaling Pathway and the Metabolic Syndrome: It Is Time for Change

**DOI:** 10.3390/ijms24054551

**Published:** 2023-02-25

**Authors:** Joseph A. M. J. L. Janssen

**Affiliations:** Department of Internal Medicine, Division of Endocrinology, Erasmus MC, 3015 GD Rotterdam, The Netherlands; j.a.m.j.l.janssen@erasmusmc.nl; Tel.: +31-06-50032421 or +31-10-7040704

**Keywords:** hyperinsulinemia, IGF-I, IGFBP-1, insulin resistance, metabolic syndrome, westernization, overnutrition, paleolithic diet, Mediterranean diet, lifestyle, prevention

## Abstract

The metabolic syndrome is a cluster of overlapping conditions resulting in an increased incidence of type 2 diabetes, cardiovascular disease, and cancer. In the last few decades, prevalence of the metabolic syndrome in the Western world has reached epidemic proportions and this is likely due to alterations in diet and the environment as well as decreased physical activity. This review discusses how the Western diet and lifestyle (Westernization) has played an important etiological role in the pathogenesis of the metabolic syndrome and its consequences by exerting negative effects on activity of the insulin–insulin-like growth factor-I (insulin–IGF-I) system. It is further proposed that interventions that normalize/reduce activity of the insulin–IGF-I system may play a key role in the prevention and treatment of the metabolic syndrome. For successful prevention, limitation, and treatment of the metabolic syndrome, the focus should be primarily on changing our diets and lifestyle in accordance with our genetic make-up, formed in adaptation to Paleolithic diets and lifestyles during a period of several million years of human evolution. Translating this insight into clinical practice, however, requires not only individual changes in our food and lifestyle, starting in pediatric populations at a very young age, but also requires fundamental changes in our current health systems and food industry. Change is needed: primary prevention of the metabolic syndrome should be made a political priority. New strategies and policies should be developed to stimulate and implement behaviors encouraging the sustainable use of healthy diets and lifestyles to prevent the metabolic syndrome before it develops.

## 1. Introduction

The metabolic syndrome (previously also called syndrome X and the insulin resistance syndrome) is characterized by a group of metabolic disturbances which are associated with an enhanced risk of type 2 diabetes, cardiovascular diseases, certain types of cancer, and mortality. Over the last fifty years, the metabolic syndrome in the Western world has reached epidemic proportions. Since the human genome has been virtually unchanged since 1950, the increased prevalence of the metabolic syndrome can be attributed to important environmental changes interacting with genes [1]. 

The “modern” Western diet is characterized by a high content of saturated fats, sugar, corn-derived fructose syrup, proteins (derived from fatty domesticated and processed meats), refined grains, low fiber content, salt, and reduced consumption of fruits and vegetables [2]. The “modern” Western diet and lifestyle may have played an important etiological role in the pathogenesis of the metabolic syndrome and its consequences. 

In this review it is argued that introduction of the “modern” Western diet and lifestyle has exerted profound negative effects on the insulin–insulin-like growth factor-I (insulin–IGF-I) system. It is further hypothesized that interventions that normalize/reduce activity of the insulin–IGF-I system might play a key role in the prevention and treatment of the metabolic syndrome. For successful prevention and treatment of the metabolic syndrome, the focus should be primarily on changing our diets in accordance with our genetic make-up, to implement long-term, sustainable behaviors, that encourage healthy lifestyles.

## 2. The Insulin–IGF-I System

The insulin–IGF system is formed by insulin, two insulin-like growth factors (IGF-I and IGF-II), and six cell-membrane signal-transducing receptors (insulin receptor-A (IR-A), insulin receptor-B (IR-B), insulin-like growth factor-I receptor (IGF-IR) and their respective hybrids): the IRs and IGF-IR can be found at the cell surface as homodimers composed of two identical alpha/beta monomers, or as heterodimers composed of two different receptor monomers (aka hybrids) [3,4,5]. The system further contains the IGF-II scavenger transmembrane protein previously known as insulin-like growth factor “receptor”-II “(IGF2R), and six IGF-binding proteins (IGFBP-1-6), several IGFBP-related proteins and IGFBP proteases, which serve to tightly regulate IGF-I and IGF-II levels/bioavailability in circulation and at the cell surface [3,4,5,6,7,8]. Insulin and IGF-I show high structural similarities. 

It has been postulated that IGF-I emerged at a very early stage in vertebrate evolution from an ancestral insulin-type gene [9]. This ancestral insulin-like gene was duplicated to form insulin and IGF-I in early (agnathan) vertebrates. Further gene duplications resulted in distinct IGF-I and IGF-II genes, which were first found during evolution in gnathosomes (jawed vertebrates) [10]. In vertebrates, insulin-like peptides predominantly functioned as growth factors promoting tissue growth and development. However, in vertebrates, this growth-promoting function has been subsumed by IGF-I and IGF-II, while insulin has acquired this new function and is primarily involved in metabolism. It has further been demonstrated that, in vertebrates, nutritional status is the key regulator of the activities of insulin and IGFs [11]. Consequently, insulin and IGFs regulate metabolism, growth and development in vertebrates in response to nutritional availability. 

IGF-I displays approximately 50% structural homology with proinsulin [12]. In humans, insulin and IGF-I are transported in plasma and delivered to cell membrane receptors [13]. In contrast to insulin, IGF-I is mainly present in the plasma bound to six different binding proteins, the so-called IGFBPs. The IGFBPs act as transport proteins and control efflux from circulating IGF-I and IGF-II [13]. IGFBPs modulate interactions between the IGFs and the IGF-IR, thereby indirectly controlling the biological actions of IGF-I and IGF-II [13]. Because of the high affinity of IGF-I for IGFBPs, <1% of circulating plasma IGF-I is in a free form, whereas in contrast, insulin is largely in a free form [14]. Free IGF-I may be the major biologically active hormonal form of IGF-I [15]. When evaluating IGF-I bioactivity in health and disease, it is therefore recommended also to measure circulating free IGF-I levels [15]. 

## 3. The Role of Insulin and IGF-I in Metabolism

Insulin and IGF-I are both complementary and involved in metabolism, growth, and in the control of essential cellular and physiological processes such as proliferation, differentiation, and body growth [16]. Although insulin is, in many respects, structurally similar to IGF-I, insulin binds with high affinity to the insulin receptor, whereas it has very low affinity for the IGF-I receptor, while the converse is true for IGF-I [17]. Binding of IGF-I to the IGF-I receptor primarily results in stimulation of proliferation and differentiation of cells [17]. Nevertheless, IGF-I also stimulates insulin-like actions (glucose and amino acid uptake, mRNA, and protein synthesis) in cells and enhances insulin sensitivity [14,18]. Most of the ability of IGF-I to enhance insulin sensitivity is mediated (indirectly) through suppression of GH, a well-known insulin antagonist [14]. In addition, IR–IGF-IR hybrids behave like IGF-IRs but show differential IGF-I vs. IGF-II stimulated activity based upon the involved IR isoforms [19,20]. 

Healthy adult liver cells show a very low expression of the IGF-IR, and the IGF-IR expression in fat cells of the adipocyte tissue is also low and tends to decrease when differentiating from pre-adipocytes to mature adipocytes [21,22]. Therefore, these latter tissues, while being extremely insulin responsive, are relatively refractory to exposure to IGF-I [14]. A complex inter-relationship exists between IGF-I, insulin, and GH [23]. Hepatic IGF-I production is stimulated by both GH and insulin, while IGF-I feeds back to suppress GH and insulin secretions [23] (see further below).

## 4. Reaven and the Metabolic Syndrome

Reaven previously suggested that the more insulin sensitive an individual, the better off (s)he is [24]. In an insulin-sensitive individual, the beta-cell is not stressed, and glucose tolerance can be maintained at low plasma insulin levels [24]. However, he also suggested that resistance to insulin-stimulated glucose uptake is present in more than 80% of individuals with impaired glucose tolerance or type 2 diabetes, and in about 25% of nonobese individuals with normal glucose tolerance [24]. Insulin resistance, by itself, is not sufficient to produce diabetes [24]. If the pancreatic beta-cells are able to overcome insulin resistance by increasing their insulin secretory response, frank type 2 diabetes will not occur. Thus, hyperinsulinemia may prevent frank diabetes in insulin-resistant individuals. However, according to Reaven, the compensatory hyperinsulinemia to overcome insulin resistance has its own price (independent from the development of diabetes) and may produce unwanted effects [24]. In 1988, Reaven hypothesized that insulin resistance and hyperinsulinemia induce the metabolic syndrome: insulin resistance and hyperinsulinemia provide a common mechanism for the development of glucose intolerance, elevated circulating triglycerides, decreased HDL-cholesterol and hypertension [24]. In the view of Reaven, glucose intolerance, elevated circulating triglycerides, decreased HDL-cholesterol, and hypertension are more likely to occur together than separately, and when they cluster, they are highly likely to be the manifestations of resistance to insulin-stimulated glucose uptake and (compensatory) hyperinsulinemia [25]. Reaven further proposed that the metabolic syndrome plays both an important role in the etiology and clinical course of three major related conditions: hypertension, coronary artery disease and type 2 diabetes [24]. The prevalence of the metabolic syndrome varies from 8% to 67% among different populations, and it has been suggested that the variety in prevalence of the metabolic syndrome and its individual components may be related to differences in genetic background, diet, levels of physical activity, population age and sex structure, levels of over- and undernutrition, body habitus, and definitions of the metabolic syndrome [26]. Insulin resistance and hyperinsulinemia appear to be central in the development of the metabolic syndrome [27,28]. However, at present it is still unclear whether there is a unifying mechanism which causes the metabolic syndrome. 

## 5. Causes of Hyperinsulinemia

In the view of Reaven (see above) and in most traditional literature, obesity is considered the main cause of insulin resistance, and insulin resistance is the abnormality leading to hyperinsulinemia. In this model, hyperinsulinemia is a compensatory response to insulin resistance. However, an increasing number of (prospective) human studies suggests an alternative scenario: in this scenario, chronic hypersecretion of insulin is the primary abnormality leading to hyperinsulinemia and precedes, initiates, and causes insulin resistance. In this new scenario, hyperinsulinemia is the first event triggering insulin resistance, obesity, the metabolic syndrome, and type 2 diabetes [29,30,31,32,33,34,35,36,37,38,39]. In addition, in subjects with normal plasma glucose concentrations, it has been found that hyperinsulinemia per se induced insulin resistance by insulin-induced downregulation of insulin receptor signaling [40,41]. This further supports the idea that hyperinsulinemia may be a primary driver of insulin resistance [41,42]. Moreover, it has been shown that gastric bypass surgery in type 2 diabetes corrects and normalizes plasma insulin levels within some days after surgery, even though there is, at that moment, no significant weight loss (yet) and insulin resistance remains high [43]. This post-operative course clearly dissociates hyperinsulinemia from insulin resistance, further supporting the idea that hyperinsulinemia is primary, rather than a consequence occurring secondary (as a compensation) to insulin resistance [43,44]. 

In a well-characterized cohort of apparently healthy adults, elevated fasting insulin at baseline was found to be an independent determinant over a 5-year period for the future development of the metabolic syndrome [45]. In addition, prospective evidence shows that—independently of obesity and body weight—hyperinsulinemia is related to the development of dyslipidemia and hypertension, suggesting that hyperinsulinemia precedes these disorders in the etiologic pathway [46,47,48]. Thus, hyperinsulinemia may indeed be an early and central feature of the cardiovascular risk of subjects with the metabolic syndrome [45]. Further support of a pathogenic role of hyperinsulinemia in the development of the metabolic syndrome was demonstrated in an animal model by Jeanrenaud et al. They showed that short-term hyperinsulinemia is a pathological driving force, which produces incipient obesity by overstimulating white adipose tissue and liver metabolic activity while concomitantly producing incipient muscle insulin resistance [49]. 

In her Banting Lecture in 2011, Barbara Corkey proposed a model in which excessive beta-cell insulin secretory responses, possibly induced by consumption of the “modern” Western diet and over-nutrition, superimposed on a susceptible genetic background and metabolic programming, may be a major cause of hyperinsulinemia, insulin resistance, obesity, and type 2 diabetes [42]. As Corkey pointed out, many aspects of the Western diet may potentially promote hyperinsulinemia: excess nutrient ingestion, artificial sweeteners, mono-oleoylglycerol, the macronutrient ratio in the food, the characteristics of the carbohydrates, proteins and fat of the food, and insufficient dietary fiber intake [50]. In the model of Barbara Corkey, hyperinsulinemia is the dominant driver of insulin resistance. In addition, in her model, insulin resistance is an adaptive response protecting muscles from chronic hyperinsulinemia-mediated nutrient excess and intracellular hyperglycemia [42,51]. 

Increased activity of incretin hormones may also play a role in the development of hyperinsulinemia. Gastric inhibitory polypeptide (GIP) is an incretin hormone secreted by the gut leading to enhanced pancreatic insulin secretion after food ingestion [52]. A significant elevation of fasting GIP was found in healthy young men after five days of ingesting a Westernized diet (i.e., a diet high in fat and calories: containing 50% extra energy, where 60% of the energy came from fat (HFHC diet) [37]. In this latter study, insulin secretion was increased and preceded the development of peripheral insulin resistance, mitochondrial dysfunction, and obesity in response to overfeeding, suggesting a direct and causal role for the Westernized diet, GIP, and hyperinsulinemia in the development of peripheral insulin resistance and obesity [37]. 

Pancreatic insulin secretion and (hepatic) insulin clearance play distinct roles in peripheral plasma insulin concentrations. Emerging evidence suggests that not only pancreatic insulin secretion, but also lower hepatic insulin clearance, may contribute to hyperinsulinemia (=increased peripheral insulin levels). A wide variation in hepatic as well as extrahepatic insulin clearance in human subjects has been reported [53]. Hepatic insulin clearance is (at least partly) a heritable trait; several genetic variants that are involved in regulating insulin clearance have been identified in Mexican Americans, an ethnic group with a high prevalence of the metabolic syndrome and diabetes [53]. After an overnight fast, hepatic first-pass insulin extraction in Afro-Americans (AAs) was found to be only one third compared to European Americans (EA) [54]. 

Bergman et al. suggested the following course of events in the development of type 2 diabetes: low(ered) hepatic insulin clearance causes peripheral hyperinsulinemia, which in turn exacerbates insulin resistance [54]. Consequently, insulin resistance will stress pancreatic beta-cells, and this may finally result in their ultimate failure and onset of frank type 2 diabetes [54]. Therefore, Bergman et al. hypothesized that low(ered) hepatic insulin clearance can be a primary cause of type 2 diabetes in high-risk individuals [54]. In their view, development of insulin resistance in muscles and fat cells is caused by of an overexposure of the (post-hepatic) peripheral tissues to endogenous insulin [54]. As a direct consequence of the peripheral hyperinsulinemia, peripheral insulin resistance develops to dampen hyperinsulinemia-mediated stimulating effects in muscles and fat cells [54]. 

Reduced hepatic insulin clearance has been reported in healthy lean subjects after only three days on a diet containing high carbohydrate (80E%), low fat (9E%), and a 75% increase in daily energy calories [55]. This suggests that a high-carbohydrate, high-calorie diet may quickly and directly change hepatic insulin clearance [55]. By contrast, energy restriction induced by Roux-en-Y gastric bypass (RYGB) surgery of obese subjects with normal glucose tolerance, and obese patients with preoperatively type 2 diabetes, increased hepatic insulin clearance (=normalized) significantly within one week after operation [56]. 

All these studies suggest that hepatic insulin clearance is a dynamic process which can be modified (within a few days) by diets with altered energy and, particularly, carbohydrate intake. In addition, all these studies point to an important role of reduced hepatic insulin clearance as the potential culprit in the development of (peripheral) hyperinsulinemia [57].

## 6. The Effects of Hyperinsulinemia on the Balance of the Insulin–GH–IGF-I Axis 

The insulin–growth hormone–IGF-I (insulin–GH–IGF-I) axis plays a pivotal role in metabolism. Nutrients stimulate the activity of the insulin–GH–IGF-I axis, and insulin, GH and IGF-I secretion are all nutritionally regulated [14,58]. Insulin secretion into the portal venous system is required for normal liver GH responsiveness and hepatic IGF-I synthesis and bioavailability [59]. Insulin is essential for growth hormone receptor expression in the liver and growth hormone-mediated hepatic IGF-I production [60]. There is also evidence that insulin may directly stimulate hepatic IGF-I expression [61]. 

Insulin (produced in the pancreas) and GH (produced in the pituitary gland) both stimulate IGF-I production in the liver, while, after secretion, IGF-I feeds back to suppress both insulin and GH secretion [60]. In healthy individuals, the insulin–GH–IGF-I axis is in balance: insulin and GH stimulate hepatic IGF-I production, whereas IGF-I secreted by the liver feeds back to suppress both insulin and GH [62,63] (Figure 1A). The typical modern Western diet can disturb this balance. Due to its continuous food intake, energy surplus, high content of sugars, corn-derived fructose syrup, saturated fats and proteins, the modern Western diet may induce hyperinsulinemia, which in turn increases IGF-I secretion [63]. Increased IGF-I subsequently induces suppression of GH secretion to lower levels than normal [64,65]. It has been found that only a few days of overeating markedly suppressed GH secretion (before any measurable weight gain) and the accompanying hyperinsulinemia is a likely mediator of this rapid reduction in GH secretion [66]. Thus, nutritionally driven hyperinsulinemia may disturb the normal balance of the insulin–GH–IGF-I axis by shifting the insulin : GH ratio towards insulin (and IGF-I) and away from GH [63] (Figure 1B). The increased insulin : GH ratio stimulates energy storage and lipid synthesis and inhibits lipid breakdown and thereby promotes obesity by promoting higher fat accumulation and lower energy expenditure [63]. Nutritionally driven disbalance of the insulin–GH–IGF-I axis may be an important etiological factor in the development of metabolic syndrome, impaired glucose tolerance and type 2 diabetes. Intriguingly, the (imbalance of the) insulin–IGF-I system is increasingly being implicated in the development of cardiovascular disease and cancer [67,68]. 

The effects of circulating IGF-I on the vasculature are largely modulated by IGFBPs, which control access of IGF-I to cell-surface IGF receptors [67]. IGFBP-1, one of the six IGFBPs, is believed to play an important role in glucose counter-regulation and the maintenance of normal glucose levels in normal and altered nutritional states [69]. In vitro and in vivo studies have demonstrated marked suppression of IGFBP-1 mRNA gene expression by insulin at the transcriptional level [70,71]. 

Circulating IGFBP-1 is inversely related to serum insulin (and IGF-I) and to pancreatic insulin secretion [72,73]. Low circulating IGFBP-1 levels may be a marker of hyperinsulinemia [73,74,75]. 

Low fasting IGFBP-1 levels are linked to an impaired glucose tolerance and increased cardiovascular risk and have also been suggested to play a role in the development of the metabolic syndrome [76,77,78,79]. 

The primary action of IGFBP-1 is to inhibit binding of IGF-I to cell membranes and thereby to reduce IGF-I bioactivity [80]. When IGFBP-1 levels are low (in case of hyperinsulinemia), IGF-I bioactivity and IGF-I-mediated effects increase and supplement the actions of insulin [69]. Conversely, when insulin levels are low (during fasting or in the late stages of type 2 diabetes), elevated IGFBP-1 levels inhibit IGF-I mediated effects [69]. 

We previously examined a cross-sectional sub-study of a random sample of 1036 elderly subjects participating in the Rotterdam Study and found that, in subjects with normal fasting glucose (NFG) and impaired fasting glucose (IFG), circulating IGF-I bioactivity (measured by a bioassay) progressively rose with increasing insulin levels and severity of insulin resistance [65]. At fasting glucose levels just below 7.0 mmol/L, fasting insulin levels and IGF-I bioactivity reached a plateau [65]. However, in subjects with diabetes (fasting glucose > 7.0 mmol/L), fasting insulin progressively decreased with higher glucose levels, whereas IGF-I bioactivity was significantly lower than in subjects with NFG and IFG [65]. Although we did not measure IGFBP-1 in our study, insulin-mediated effects on IGFBP-1 may have been responsible for the observed changes in circulating IGF-I bioactivity in subjects with NFG, IFG, and diabetes (see below). 

It is commonly accepted that during the natural course from normoglycemia to frank type 2 diabetes, insulin secretion follows the pattern of an inverted U, also termed ”Starling’s curve of the pancreas” (Figure 2A). 

Although cause and effect may be difficult to assess within a cross-sectional study, the above-mentioned sub-study embedded in the Rotterdam Study showed that IGF-I bioactivity also follows the pattern of an inverted U during the natural course from NFG or IFG to diabetes, suggesting also a “Starling Curve” for IGF-I bioactivity (Figure 2B). The inverted U pattern of IGF-I bioactivity is probably caused by direct modulating effects of insulin (and IGFBP-1) on IGF-I bioactivity during the natural course from fasting normoglycemia to frank diabetes (Figure 2B). In line with our findings, Cruickshank et al. previously found a U-shaped relationship between insulin and IGFBP-1 during the natural course from normoglycemia to frank type 2 diabetes in a cross-sectional study among Europeans and Pakistanis: IGFBP-1 concentrations were lower in those with impaired compared with normal glucose tolerance, whereas IGFBP-1 rose again in those with diabetes, probably reflecting insulin arriving at the liver, the principal site of IGFBP-1 synthesis (Figure 2C) [81].

## 7. Prevalence of the Metabolic Syndrome among Ethnic Immigrants 

Important differences in the prevalence of the metabolic syndrome can be found among ethnic immigrant groups [82]. As already discussed, susceptibility to the metabolic syndrome can predominantly be attributed to environmental factors [82]. Environmental factors may also have profound effects on the insulin–IGF-I system and its relationship with related metabolic variables [83]. Previously, marked differences in parameters of the insulin–IGF-I system were observed in a cross-sectional population-based community study of Gujaratis, who all originated from the same Indian village (Navsari) [83]. One group of Gujaratis had migrated to Sandwell, UK, and were the first so-called first-generation migrants. The other group of Gujaratis was still living in their hometown of Navsari in Gujarat in India. Body mass index (BMI), waist to hip ratio (WHR), and systolic and diastolic blood pressure were observed to be substantially higher in the Gujaratis living in the UK, although both groups were of comparable age [83] (Table 1). Circulating fasting insulin, IGF-I, and IGFBP-3 were significantly higher, while fasting IGFBP-1 was significantly lower in Sandwell Gujarati men and women than in Navsari Gujaratis [83] (Table 1 and Figure 3). As expected, there was a highly significant inverse relationship between IGFBP-1 and fasting insulin concentrations [83]. IGFBP-1 at both sites correlated negatively with BMI, WHR, diastolic blood pressure, serum triglycerides, fasting insulin, and 2 h insulin [83]. When dietary macronutrient intake was compared between Sandwell Gujaratis and Navsari Gujaratis, total energy, fat, protein, and carbohydrate intake were significantly higher in the Sandwell Gujaratis, and these differences were more pronounced in men than in women (Table 1) [84]. In addition, at both sites, IGF-I correlated positively with total energy and total fat intake [84]. 

Portal blood insulin levels are thought to be amongst the most potent regulators of IGFBP-1 secretion from the liver [85]. As discussed above, fasting IGFBP-1 levels can be used as a marker of insulin secretion in healthy subjects. In general, an inverse correlation between fasting levels of IGFBP-1 and actual 24 h insulin secretion can be observed [78]. Only in Indian Gujaratis did IGFBP-1 show an inverse relationship with total energy and fat intake, whereas such a relationship was not observed in Sandwell Gujaratis [84]. The absence of change in IGFBP-1 levels in relation to increased total energy and fat intake in Sandwell Gujaratis was argued to be the consequence of already maximally insulin-mediated suppressed IGFBP-1 baseline levels. A further change in diet could therefore make no or only very little change (decreases) to circulating IGFBP-1 [84]. 

Thus, in these genetically similar groups, migration to the UK and adoption of a different diet was associated with marked changes in the insulin–IGF-I system, suggesting that environmental factors, particularly (quantity and types of) macronutrient intake, profoundly modulate serum concentrations and actions of the insulin–IGF-I system [84]. 

## 8. Insulin Levels in Traditional Populations Not Using a Western Diet

Traditional ethnic groups, in general, seem to be more prone, not less, to developing obesity, diabetes, and cardiovascular disease after adopting a Western lifestyle. On the tropical island of Kitavi, Papua Guinea, live the Kitavans, the original inhabitants of Kitavi. In Kitava, stroke and ischemic heart disease are absent or rare, and body mass index (BMI) and diastolic blood pressure are low in Kitavans [86]. Since the intake of Western food is negligible in the traditional Kitavans, they provide, at this moment, one of the last opportunities on our planet to study a group of humans who are not influenced by Western dietary habits [86]. In contrast to a Western diet, the traditional Kitavan diet has a low glycemic index and is low in energy and fat, but nevertheless has high satiety power [86]. Fasting insulin and glucose levels in Kitavans were lower compared with randomly selected Swedish controls from the MONICA study [86]. For example, mean insulin concentration in 50- to 74-year-old Kitavans was only 50% of that in Swedish subjects, despite a normal glucose tolerance [86]. Even more interesting, serum insulin levels decreased with age in Kitavans; in contrast, they increased in Swedish subjects over 50 years of age [86]. The age-related decrease of serum insulin in Kitavans was lost after adjustment for midarm circumference, a proxy for muscle mass in this lean and insulin-sensitive population [86]. Because low fasting serum insulin levels in Kitavans did decrease with older age, and insulin is needed for glucose uptake into muscle cells, it was concluded that low fasting insulin levels in lean populations with a low-risk for cardiovascular disease—such as the Kitavans—is probably a marker of (loss of) muscle mass during aging rather than a marker of insulin sensitivity [86]. In addition, it was also concluded that the observed increased fasting insulin levels with age in Swedish subjects over 50 years of age are probably not part of normal aging, but an untoward effect of a Western diet and lifestyle on insulin secretion and sensitivity [86]. Effects of a Western diet and lifestyle on fasting insulin levels may, in this respect, show analogy to the typical age-related increase in waist circumference and blood pressure found in subjects following a Western diet and lifestyle [86]. 

In contrast to a Western diet, the traditional Kitavan diet has a low glycemic index and is low in energy and fat [86]. Interestingly, fat is responsible for only about 20% of the total daily calories in the traditional Kitavan diet, whereas carbohydrates provide 70% of the total daily calorie intake [86]. The low fasting insulin levels observed in Kitavans using the traditional Kitavan diet suggests that quality rather than quantity of carbohydrates is probably more important for the level of pancreatic insulin secretion; carbohydrate-rich foods with a low glycemic index and a high nutrient density, which are a typical part of the traditional Kitavan diet, appears in this respect to be better than refined cereals and sugar [86]. 

## 9. Low(er) Activity of Insulin/IGF-I Signaling Pathway Protects against Type 2 Diabetes and Cancer

Laron syndrome is an autosomal recessive disorder characterized by a lack of IGF-I production in response to GH [87]. It is caused by inherited GH receptor mutations which result in varying severity of GH insensitivity [87]. Individuals with the classic Laron syndrome present with short stature, obesity, low blood sugar, and congenital IGF-I deficiency (with low serum IGF-I) with decreased insulin/IGF-I signaling activity despite elevated basal serum GH [87]. Guevera-Aguire et al. found in individuals with Laron syndrome, in contrast to their healthy relatives with normal insulin/IGF-I signaling, a significant reduction in pro-aging signaling, cancer, and type 2 diabetes [88]. Serum from subjects with the Laron syndrome induced in vitro a reduced number of DNA breaks but increased apoptosis in human mammary epithelial cells treated by hydrogen peroxide [88]. Moreover, serum from subjects with the Laron syndrome also caused reduced expression of rat sarcoma virus (RAS), PKA (protein kinase A), and mTOR (target of rapamycin) and up-regulation of superoxide dismutase 2 in treated cells [88]. All these changes promote normal cellular protection and life-span extension in model organisms and provide a possible explanation for the observed low incidence of cancer in subjects with the Laron syndrome in the study by Guevera-Aguire et al. [88]. Moreover, individuals with the Laron syndrome showed reduced insulin concentrations (1.4 μU/mL versus 4.4 μU/mL in unaffected relatives) and a very low HOMA-IR (homeostatic model assessment-insulin resistance) index (0.34 versus 0.96 in unaffected relatives), indicating that higher insulin sensitivity could provide a possible (alternative) mechanism explaining the reduced prevalence of type 2 diabetes and cancer observed in subjects with the Laron syndrome [88]. Thus Laron syndrome, an experiment of nature, showed that in humans, low(er) activity of the insulin/IGF-I signaling pathway and high insulin sensitivity may play a crucial role in the protection from diseases typically related to Western civilization, such as type 2 diabetes and cancer [89]. 

In contrast to the Laron syndrome, active acromegaly shows increased GH secretion and IGF-I activity. Main metabolic consequences of active acromegaly are hyperinsulinemia and insulin resistance despite a lean phenotype, and this results in an increased risk of type 2 diabetes [90,91]. Control of active acromegaly reduces hyperinsulinemia, improves insulin sensitivity, increases fat mass and glucose homeostasis in most, but not in all, studies [90]. Underlying pathophysiological mechanisms for the hyperinsulinemia, insulin resistance, and prediabetes/diabetes in active acromegaly, are GH-mediated increased lipolysis, reduced peripheral glucose utilization, and enhanced gluconeogenesis by the insulin-antagonizing effects of GH [90,91]. The sustained stimulation of lipolysis mediated by GH plays not only a major role in the development of hyperinsulinemia, insulin resistance and prediabetes/diabetes, but also in the reduction of lipid accumulation and the development of disadvantageous metabolic changes and comorbidities, such as hypertension, cardiovascular diseases, malignant neoplasms, and the resulting decreased life expectancy [91]. 

Metabolic syndrome and untreated adult-onset growth hormone deficiency share several key features, including abdominal visceral obesity, insulin resistance, impaired glucose tolerance and/or type 2 diabetes mellitus, hypertension, hypertriglyceridemia, and decreased high-density lipoprotein (HDL)-cholesterol levels [92]. It has been previously found that GH replacement therapy had favorable effects on separate components of the metabolic syndrome, apart from a possible deterioration of insulin sensitivity and glycemic control, particularly in patients with elevated body mass index [93,94,95,96]. Despite favorable changes in some metabolic abnormalities seen in several controlled studies of patients with growth hormone deficiency during GH replacement therapy, prevalence of the metabolic syndrome (using a composite measure for the metabolic syndrome as a major outcome) remained unchanged or even increased during long-term GH replacement therapy [97,98,99]. It therefore can be concluded that GH replacement therapy does not reduce metabolic syndrome risk in subjects with adult-onset growth hormone deficiency. 

## 10. The Activity of the Insulin–IGF-I Signaling Pathway and Longevity

Disruption of genes in the insulin–IGF-I signaling pathway that share similarities with those in humans can significantly extend life span in diverse species, including yeast, worms, fruit flies, and rodents [100]. It has therefore been suggested that reducing the activity of the insulin–IGF-I signaling pathway plays a key role in delayed aging and prolonged longevity [101]. A point to emphasize here, and in favor of this suggestion, is that all long-lived mutants, ranging from yeast to mice, share some important phenotypic characteristics, including reduced insulin signaling, enhanced insulin sensitivity, and reduced IGF-I plasma levels [100]. 

Laboratory animals fed ad libitum have relatively low levels of physical activity and therefore show similarities to humans with a Western (sedentary) lifestyle, who are at high risk for hyperinsulinemia, obesity, and insulin resistance [102]. Caloric restriction counteracts the general trend for laboratory animals to progressively increase fat mass during aging [102]. Restriction of the number of calories consumed lowers the incidence of age-related loss of functions and disease and increases life span in a wide variety of animals [102,103]. It has been further suggested that many of the beneficial effects of caloric restriction on lifespan are mediated, at least in part, by down-regulation of the insulin/IGF-I signaling pathway activity [104]. In support of this latter option is the observation that caloric restriction in rodents decreases activity of the insulin/IGF-I signaling cascade and postpones or attenuates cancer, immunosenescence, and inflammation without irreversible side effects [102]. 

While chronic hypernutrition may lead to detrimental consequences, chronic undernutrition/malnutrition should also be avoided during dietary interventions since this may evoke starvation effects, decreasing health and negatively impacting lifespan [105].

Nutrients and insulin both activate the mTOR pathway and this pathway is involved in cellular senescence and age-related diseases [104,106,107]. Overnutrition increases insulin secretion, increases IGF-I bioactivity and hyperactivates the mTOR pathway [107] (Figure 4A). Hyperactivation of the mTOR pathway induces insulin receptor resistance which blocks insulin-mediated glucose uptake and results in elevated glucose levels [107]. The elevated glucose levels induce a further increase in insulin secretion, which will, in turn, further deteriorate insulin sensitivity [104]. In contrast, lifespan-extending caloric restriction without malnutrition, decreases the activity of the mTOR pathway: this improves insulin receptor sensitivity and secondarily reduces plasma glucose levels, insulin levels and IGF-I bioactivity [104,106] (Figure 4B). 

Healthy centenarians are often considered the best living model of successful human aging and therefore often used to study healthy longevity [108]. Interestingly, it has been found that in healthy centenarians, fasting insulin levels, glucose tolerance, glucose-stimulated insulin secretion, and insulin sensitivity, are low and comparable to those found in healthy subjects aged 50 years or younger [109]. Thus healthy centenarians have preserved insulin actions comparable to subjects aged 50 years or younger, again suggesting that hyperinsulinemia, age-related insulin resistance and reduced insulin actions are not an obligatory finding in the elderly [110] (see also above paragraph on Insulin levels in traditional populations not using a Western diet about the Kitavans). 

It is still unclear at present why insulin actions and insulin sensitivity remain preserved in healthy centenarians. Parr previously hypothesized that low(er) insulin levels due to increased insulin sensitivity might provide healthy centenarians with a well-balanced insulin–GH–IGF-I axis, which may induce a slower loss of physiologic reserves, thereby permitting better conditions for a longer life span [108]. In addition, healthy centenarians usually show significantly less oxidative stress and greater plasma antioxidant defenses than aged subjects [111]. It has further been clearly shown that oxidative stress is a precursor to insulin resistance [112]. Due to the vicious circle occurring between insulin and oxidative stress, it has been proposed that low insulin (and IGF-I) levels with preserved insulin sensitivity are important factors responsible for the observed higher insulin sensitivity and lower oxidative stress in healthy centenarians compared with aged subjects [110]. 

The remodeling theory of aging suggests that healthy centenarians are probably not “the best” but “the best adapted” to (potentially damaging) environmental factors they have encountered during their lifetimes [113]. Preserved insulin action and sensitivity within an individual during aging may merely be the expression of a successful adaptation of the body during his/her lifetime. Conversely, if this adaptation of the body is less successful, age-related insulin resistance and beta-cell dysfunction may develop, which shortens life expectancy [113]. 

## 11. Insulin Sensitivity Improves after Temporary Reversion of Traditional Hunter-Gatherer Lifestyle

As discussed, Western lifestyle factors (diet and a low level of physical activity) may play an important role in the worldwide prevalence of the metabolic syndrome and type 2 diabetes. Many populations have been subjected to Westernization during the last century and some have appeared particularly vulnerable to its impact. As a direct consequence, extremely high prevalence rates of obesity, the metabolic syndrome, and type 2 diabetes have developed in Pima Indians, Nauruans, multi-ethnic groups in Mauritius, and Australian Aborigines after Westernization [114,115,116,117]. Australian Aborigines represent a good example of a population that has developed a high prevalence of the metabolic syndrome and type 2 diabetes after introduction of a Western lifestyle. In common with many other populations at high risk for diabetes, Australian Aborigines showed hyperinsulinemia (both fasting and in response to oral glucose), obesity (with an android pattern of fat distribution), impaired glucose tolerance, hypertriglyceridemia and hypertension, when they adopted a Western lifestyle [117,118,119]. The changes in lifestyle in Australian Aborigines can be characterized by major changes in their diet and physical activity in conjunction with the disruption of traditional and social structures [118]. However, when the Australian Aborigines still lived as hunter-gatherers, the (potential) predisposition to obesity (an android pattern of fat distribution), metabolic syndrome, and type 2 diabetes was not expressed. Their traditional diet was low in energy, high in protein, low in carbohydrate and fat, while most carbohydrates were derived from fiber-rich, slowly digested foods, and polyunsaturated fats [118]. This traditional diet minimalized hyperinsulinemia postprandially. In contrast, when the Australian Aborigines left their hunter-gatherer lifestyle and switched to a Western diet, there was an unlimited availability of food and the diet became very rich in sugar and fat, very high in energy and low in protein. These dietary changes produced hyperglycemia and stimulated a high insulin response [120]. 

Interestingly, when ten diabetic Australian Aborigines were reverted to their hunter-gatherer lifestyle for only seven weeks, dramatic changes were observed [121]. Food intake revealed a low-energy intake (1200 kcal/person/day) [121]. The diet was low in total fat (13%) due to the very low-fat content of wild animals [121]. All subjects, most of whom were overweight initially, lost weight (mean weight baseline, 81.9 kg; after seven weeks, 73.8 kg) [121]. Mean fasting glucose fell (mean glucose baseline, 11.6 mmol/L; after seven weeks, 6.6 mmol/L) and glucose tolerance improved significantly. In addition, mean fasting insulin fell (mean fasting insulin, 23 mU/L at baseline; after seven weeks, 12 mU/L); after seven weeks the insulin secretory response after an oral glucose load was improved, while the hypertriglyceridemia normalized [121]. Although insulin sensitivity was not directly measured in this study, the reduction in fasting insulin and the improvement of glucose tolerance suggest that it was improved. In addition, weight loss and increased physical activity, observed during reversion to the traditional hunter-gatherer lifestyle, are well-known known factors that improve insulin sensitivity. 

## 12. How to Halt the Negative Impact of the Western Lifestyle on the Insulin/IGF-I System and the Prevalence of the Metabolic Syndrome

The increasing prevalence of the metabolic syndrome seems primarily driven by changes in diet and increasingly sedentary lifestyles [122]. Due to the Western dietary pattern of frequent snacking and frequent consumption of sucrose-containing soft drinks, insulin levels are elevated most of the day [123,124]. The changes in dietary habits, adopted by the Western world over the past 100 years, appear to have made an important contribution to the increasing prevalence of the metabolic syndrome and its consequences—coronary artery disease, hypertension, diabetes, and some cancers. These conditions have only emerged in the past century but were virtually absent in hunter-gatherer populations following a traditional hunter-gatherer (or paleolithic) diet and lifestyle [125]. The traditional hunter-gatherer diet was the typical diet that human beings had adapted to consume during a long period of our evolution [125]. The human genetic constitution has formed over a period of several million years and changed relatively little since the appearance of human sapiens about 40,000 years ago [1]. Significant changes in the Western diet started after the Industrial Revolution in the 19th century and thus occurred far too recently on the evolutionary time scale for the human genome to adjust [125]. A period of only a few hundred years is considered too short to induce an adequate adaptation of the human genome to significant and recent changes in the environment [125,126]. Consequently, the genetic make-up of humans is still, at present, best adapted to the low-glycemic and low-insulinemic hunter-gatherer (paleolitic) diet, and it therefore has been suggested that, although we are people living in the 21st century, genetically we are still citizens of the Paleolithic era [123,127]. 

As discussed previously, the “modern” Western diet and lifestyle, by exerting negative (i.e., stimulating) effects on the activity of the insulin–IGF-I system, may play an important etiological role in the pathogenesis of the metabolic syndrome. Interventions that normalize/reduce activity of the insulin–IGF-I system might therefore play a key role in the prevention and treatment of the metabolic syndrome and its consequences. 

Primary prevention of the metabolic syndrome (i.e., before it starts) is probably the only effective and cost-effective approach, counterbalancing the environmental roots of the increased prevalence of the metabolic syndrome. Some advocate that we should turn to the paleolithic diet and lifestyle when seeking solutions for the increased prevalence of hyperinsulinemia, insulin resistance, obesity, and type 2 diabetes, which have emerged in many populations worldwide after switching to the Westernized way of life [125]. 

Low insulin-sensitizing activity of the traditional hunter-gatherer diet (containing very low fat (10–15% of total calories)) has been repeatedly demonstrated to exert a markedly favorable impact on insulin levels, such that diurnal insulin secretion is down-regulated [128]. In addition, the traditional hunter-gatherer diet contains a relatively low protein content with the near absence of ”high-quality” animal protein, and this may decrease hepatic synthesis of IGF-I while increasing that of its functional antagonist IGFBP-1, and also possibly blunt diurnal insulin secretion [128]. Moreover, although protein per se has a modest impact on insulin secretion, it has been suggested that it can markedly potentiate the insulin response when co-ingested with carbohydrates [129]. Thus, a diet with a relatively low protein may also help to prevent hyperinsulinemia. 

Interestingly, the hunter-gatherer diet shows similarities with the traditional Mediterranean diet in many respects (Table 2) [127]. The Mediterranean diet is associated with better cardiovascular health outcomes, including clinically meaningful reductions in rates of coronary heart disease, ischemic stroke, and total cardiovascular disease [130]. The Mediterranean diet has also been shown to decrease the risk of the metabolic syndrome. A meta-analysis of 50 studies and 534,906 individuals showed that the greater the adherence to the Mediterranean diet, the greater the reversion of the metabolic syndrome and its components [130]. In the Attica study, it was found that adherence to the Mediterranean diet was linked to improved fasting glucose homeostasis, insulin levels, and a better insulin resistance index (HOMA) in both normoglycemic individuals and diabetic participants [131]. In addition, participants who had a high score of Mediterranean diet adherence showed 15% lower basal glucose and insulin, and a 27% increase in the HOMA index, a measure for insulin receptor sensitivity [131]. Recent studies in experimental animal models and in humans also yielded encouraging results for insulin-sensitizing nutritional supplements derived from Mediterranean diet nutrients [132]. The typical Mediterranean diet is characterized by low-glycemic index foods, a low number of dairy products, and a high amount of antioxidant and anti-inflammatory nutrients that also may be able to modulate activity of the insulin/IGF-I signaling pathway [133]. Data from randomized clinical trials showed that individuals randomized to a Mediterranean diet not only lost a significant amount of body weight, but also experienced a substantial reduction in fasting glucose, C-peptide (a measure for endogenous insulin secretion), and in the area under the curve for insulin after a glucose tolerance test, but also a significant increase of serum levels of IGFBP-1, -2, and a non-significant increase of growth hormone [134]. Although free IGF-I was not measured, the observed increases in IGFBP-1 and -2 most likely caused a reduction in circulating free IGF-I, which reflects, as discussed above, the major biologically active hormonal form of IGF-I [15]. Thus, these data suggest that the Mediterranean diet, by acting on the insulin/IGF-I signaling pathway, may have a protective role in the pathogenesis of the metabolic syndrome. 

Lifestyle changes such as regular and increased daily physical activity (at least one hour per day), not smoking, reducing sodium intake, limiting alcohol intake, and maintaining a healthy body weight may have additional effects and thereby modify the incidence and prevalence of the metabolic syndrome and its consequences. 

In conclusion, the Western diet and lifestyle (Westernization), by increasing activity of the insulin–IGF-I system, may play an important etiological role in the pathogenesis of the metabolic syndrome. Interventions that normalize or reduce the activity of the insulin–IGF-I system might play a key role in the prevention and treatment of the metabolic syndrome and its consequences. 

Modifications of our diets and lifestyle, in accordance with our genetic constitution—formed in adaptation to Paleolithic diets and lifestyles during a period of several million years of human evolution—may help prevent or limit the development of the metabolic syndrome. Translating this insight into clinical practice, however, requires not only individual changes in our food and lifestyle and the early start and adoption of healthy habits at a young age in pediatric populations, but also requires fundamental changes in our health system and food industry. Change is needed. Primary prevention of the metabolic syndrome should be made a political priority. New strategies and policies should be developed to implement behaviors that encourage the sustainable use of healthy diets and lifestyles to prevent the metabolic syndrome before it develops.

## Figures and Tables

**Figure 1 ijms-24-04551-f001:**
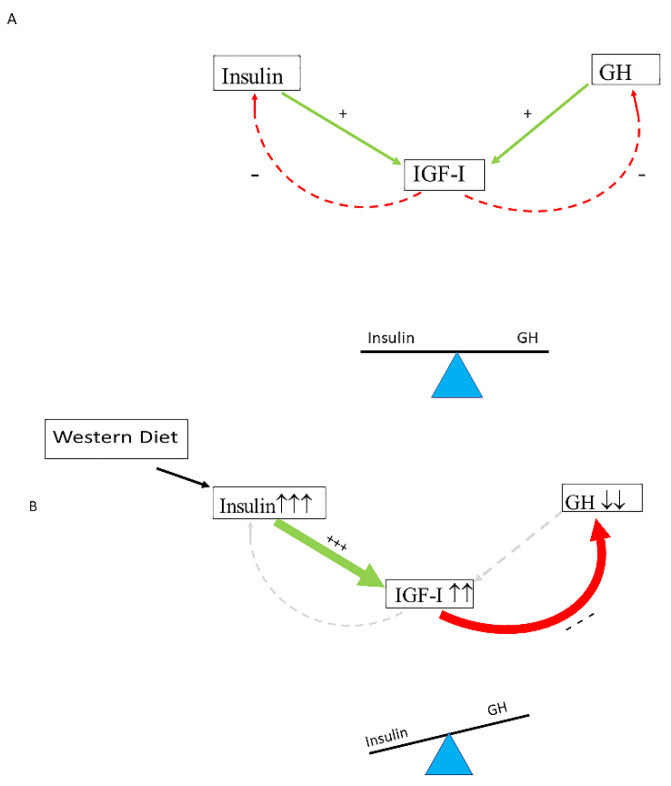
Nutritionally driven hyperinsulinemia can disturb the normal balance of the Insulin–GH–IGF-I axis. (**A**) In healthy individuals, the insulin–GH–IGF-I axis is in balance: insulin and GH stimulate hepatic IGF-I production, whereas IGF-I secreted by the liver feeds back to suppress both insulin and GH. (**B**) The modern Western diet may induce hyperinsulinemia, which in turn increases IGF-I secretion. Increased IGF-I levels subsequently induce suppression of GH secretion to lower levels than normal. Only a few days of overeating may markedly suppress GH secretion before any measurable weight gain, and it has been suggested that, in these circumstances, the accompanying hyperinsulinemia is a likely mediator of this rapid reduction in GH secretion. Consequently, a shift of the insulin : GH ratio towards insulin (and IGF-I) and away from GH will occur. The higher insulin : GH ratio lowers energy expenditure and induces fat accumulation, thereby promoting energy storage and lipid synthesis and hindering lipid breakdown. This will promote obesity because of higher fat accumulation and lower energy expenditure (see text for more details on how the Western diet may influence the balance of the insulin–GH axis). +/green: stimulating; −/red: inhibiting. +++ strong stimulation, --- strong inhibition, ↑↑↑ marked increase, ↓↓ moderately decrease.

**Figure 2 ijms-24-04551-f002:**
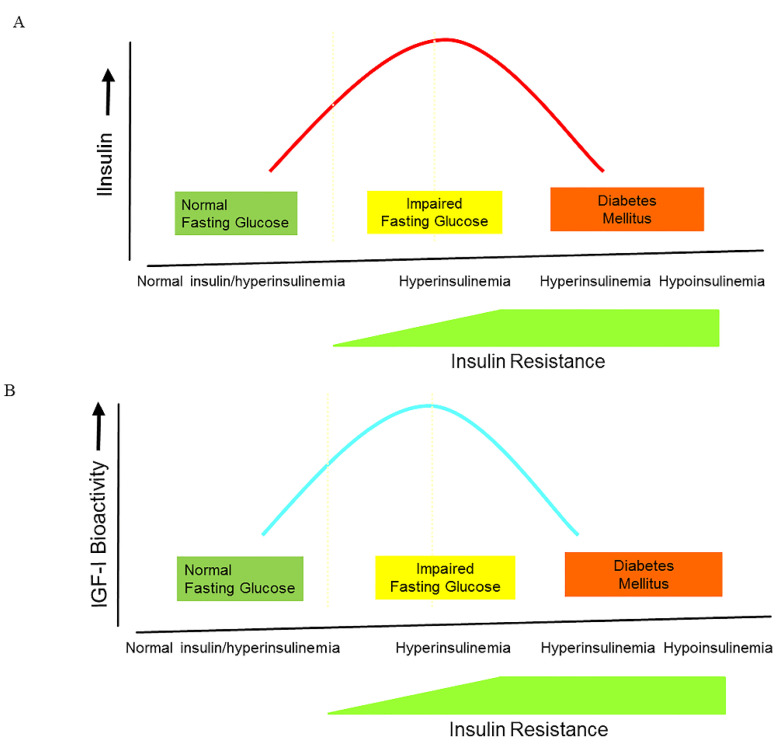
The changes in insulin, IGF-I bioactivity, and IGFBP-1 levels during the natural course from normoglycemia to impaired glucose intolerance and frank type 2 diabetes. (**A**) During the natural course from normoglycemia to frank type 2 diabetes, insulin secretion follows the pattern of an inverted U, also termed ”Starling’s curve of the pancreas”. (**B**) During the natural course from normoglycemia to frank type 2 diabetes, IGF-I bioactivity follows the pattern of an inverted U, suggesting that there is also a “Starling Curve” for IGF-I bioactivity. * (**C**) There is a U-shaped relationship between insulin and IGFBP-1 during the natural course from normoglycemia to frank type 2 diabetes **. (Modified from * [65]; ** [81]).

**Figure 3 ijms-24-04551-f003:**
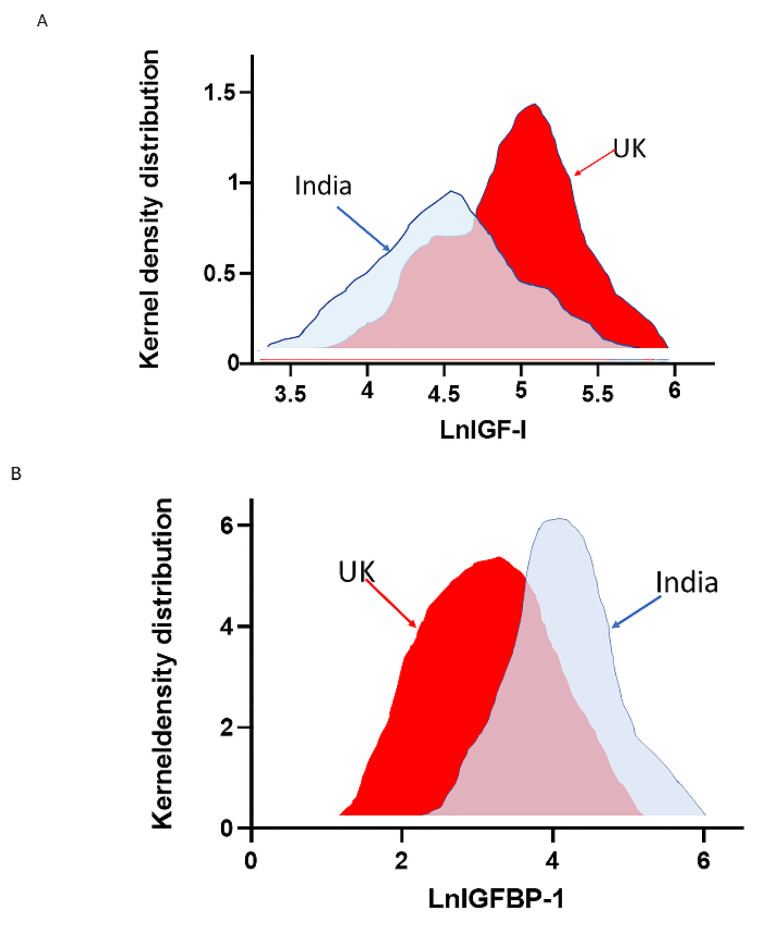
Shift in distribution of IGF-I (**A**) and IGFBP-1 (**B**) by migrant status. IGF-I was higher in UK Gujarati men and women. Conversely, fasting IGFBP-1 (age-adjusted) was significantly lower in Sandwell Gujarati men and women than in Navsari Gujaratis. Blue = Navsari group, Red = Sandwell group. (Modified from [83]).

**Figure 4 ijms-24-04551-f004:**
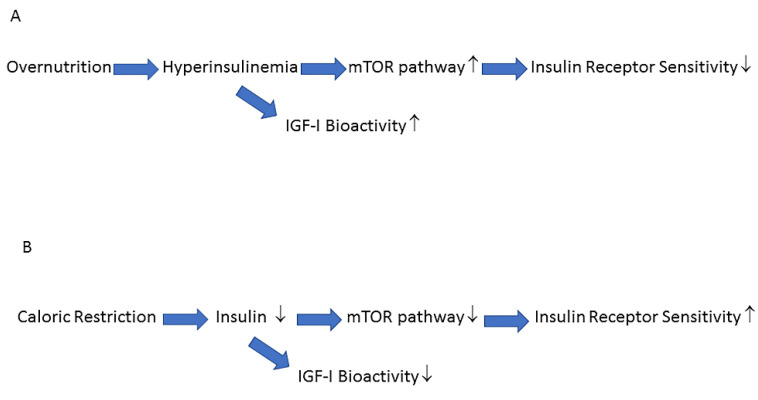
Nutrients and insulin activate the target of rapamycin (mTOR) pathway and increase IGF-I bioactivity. (**A**) Overnutrition increases insulin levels, IGF-I bioactivity and hyperactivates the mTOR pathway. Hyperactivation of the mTOR pathway induces insulin resistance and blocks insulin-mediated effects on glucose metabolism, resulting in elevated plasma glucose levels. (**B**) Caloric restriction (without malnutrition) reduces insulin levels, IGF-I bioactivity, and the activity of the mTOR pathway. The reduced activity of the mTOR pathway improves insulin receptor sensitivity and insulin-mediated effects on glucose metabolism, resulting in lower plasma glucose levels. ↑ increased, ↓ decreased.

**Table 1 ijms-24-04551-t001:** Characteristics, differences in nutritional intake and parameters of the insulin–IGF-system between Gujarati Indians living in Navsari (India) and Sandwell (UK) #.

	Men		Women	
	Navsari	Sandwell	Navsari	Sandwell
Age (years)	49.1	49.0	48.5	49.2
Height (m)	1.64	1.67 *	1.52	1.53
Body Mass Index (kg/m^2^)	21.0	25.9 **	20.8	26.6 **
Waist-to-hip ratio	0.87	0.92 **	0.79	0.82 *
Systolic BP (mm Hg)	122	134	111	121
Diastolic BP (mm Hg)	75	84	69	75
Total energy intake ^a^ (kcal/day)	1478	2221 **	1260	1720
Total fat intake ^a^ (g/day)	55.1	97.2 **	45.9	74.3 **
Total protein intake ^a^ (g/day)	52.1	78.7 **	38.4	54.7 **
Carbohydrate intake ^a^ (g/day)	177.5	268.4 **	164.7	226.5 **
IGF-I (ng/L)	113.6	154.6 **	92.0	132.8 **
IGFBP-1 ^a^ (µg/L)	45.9	18.7 **	41.9	28.1 *
Fasting plasma glucose (mmol/L)	5.4	5.2	5.3	5.2
Fasting insulin ^a^ (pmol/L)	8.2	10.2 *	9.0	9.3

Values are arithmetic means except ^a^ which show medians. * *p* < 0.05; ** *p* < 0.001; Navsari vs. Sandwell. BP = blood pressure. # Modified from [83,84].

**Table 2 ijms-24-04551-t002:** Comparison of Hunter-gatherer diet and Traditional Mediterranean diet *.

	Hunter-Gatherer Diet	Traditional Mediterranean Diet
Carbohydrates (%)	Moderate (22–40)	Moderate (50)
Total Fat (%)	Moderate (28–47)	Moderate (30)
Monosaturated Fat	High	High
Polyunsaturated fat	Moderate	Moderate
Omega-3-fat	High	High
Total Fiber	High	High
Fruits and vegetables	High	High
Nuts and seeds	Moderate	Moderate
Refined sugars	Low	Low
Glycemic load	Low	Low
Saturated Fat	Moderate	Low
Protein (%)	High (19–35)	Moderate (16–23)
Salt	Low	Moderate

Note the striking similarities between the Hunter-gatherer diet and the Traditional Mediterranean diet (upper part of the Table). and the differences in saturated fat, protein percentage and salt between the Hunter-gatherer diet and the Traditional Mediterranean diet (bottom of the Table) * Modified from: [127].

## Data Availability

Not applicable.

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
