# Peer review of "The Impact of Westernization on the Insulin/IGF-I Signaling Pathway and the Metabolic Syndrome: It Is Time for Change"

_ijms, 2023, doi:10.3390/ijms24054551_

Round 1
Reviewer 1 Report
This manuscript is a very well-written, comprehensive review focusing on the GH-IGF1, insulin axis in the pathogenesis of metabolic syndrome. The author first summarizes the physiological function and regulation of GH-IGF1 axis and insulin. Then the author discusses conceptual evolution and current understanding of metabolic syndrome, especially emphasizing the potential causal role of hyperinsulinemia (and accompanying changes in IGF1/IGFBP1), which may come first in the pathogenesis. After that, the author discusses the potential causality of western lifestyle, especially diet, in metabolic syndrome through an insightful review of epidemiological studies. Finally, the author argues preventive approaches to metabolic syndrome, in which he emphasizes the importance of lifestyle interventions and social/political actions.
The author's arguments are based on a broad range of knowledge, from molecular cellular biology to epidemiological or even historical literatures, which makes this manuscript valuable.
I would raise a couple of points that might further strengthen arguments.
1. In section 2, around lines 55-57, the author may add some more specific descriptions of the evolution in insulin/IGF-1/2 genes. A brief phylogenetic description of those anabolic hormones related to behavioral or dietary changes might strengthen the argument.
2. In sections 6 or 9, the author might extend the discussion on some other GH-IGF-1-related disorders. For example, by comparing Laron syndrome with GH deficiency, he might argue that while low IGF-I may be protective against metabolic syndrome, low GH is probably not protective because adult GHD is related to metabolic syndrome probably reflecting the protein-sparing effect of GH. Also, acromegaly can also be mentioned since it is a mirror image in terms of IGF1 levels and life expectancy.
3. While I generally agree with the author's argument on the western lifestyle in sections 9-12, I would suggest that the author may mention that the energy intake is not "the lower the better." For example, patients with anorexia nervosa typically show very high GH with suppressed IGF-1, but of course, this condition is not associated with longevity. Also, we probably do not know at what age we can safely start calorie restriction.
Minor
line 402, type 2 "diabetes"
Author Response
Reviewer 1
The author's arguments are based on a broad range of knowledge, from molecular cellular biology to epidemiological or even historical literatures, which makes this manuscript valuable.
Thank you for your kind words and the careful review of the manuscript.
We have completed our revisions based on the Reviewer‘s helpful comments, and below we indicate point by point the changes we made to the manuscript.
Answers to comments of Reviewer 1:
1. In section 2, around lines 55-57, the author may add some more specific descriptions of the evolution in insulin/IGF-1/2 genes. A brief phylogenetic description of those anabolic hormones related to behavioral or dietary changes might strengthen the argument.
A: Thank you very much for this very good suggestion. In the revised manuscript a brief phylogenetic description of those anabolic hormones was added (See revised manuscript page 2, lines 62-72).
2. In sections 6 or 9, the author might extend the discussion on some other GH-IGF-1-related disorders. For example, by comparing Laron syndrome with GH deficiency, he might argue that while low IGF-I may be protective against metabolic syndrome, low GH is probably not protective because adult GHD is related to metabolic syndrome probably reflecting the protein-sparing effect of GH. Also, acromegaly can also be mentioned since it is a mirror image in terms of IGF1 levels and life expectancy.
A: Thank you very much for this suggestion.
In the revised manuscript we added information comparing Laron syndrome and acromegaly and comparing the metabolic syndrome and untreated adult-onset growth hormone deficiency. See revised manuscript (See revised manuscript page 12, lines 464-477 & lines 478-490).
3. While I generally agree with the author's argument on the western lifestyle in sections 9-12, I would suggest that the author may mention that the energy intake is not "the lower the better." For example, patients with anorexia nervosa typically show very high GH with suppressed IGF-1, but of course, this condition is not associated with longevity. Also, we probably do not know at what age we can safely start calorie restriction.
A] Thank you for this suggestion. To meet Reviewer 1 the following information was added to the revised manuscript: While chronic hypernutrition may lead to detrimental consequences, chronic under-nutrition/malnutrition should also be avoided during dietary interventions since this may evoke starvation effects, decreasing health and negatively impacting lifespan [105]. (See revised manuscript page 13, lines 513-515).
Minor
line 402, type 2 "diabetes"
A] This was modified in the revised manuscript (page 10, line 406).
Reviewer 2 Report
Overall: Major revision required
Review author specific comments:
In the review entitled “The impact of Westernization on the insulin/IGF-I signaling pathway and the Metabolic Syndrome: It is Time for Change”, the author summarizes a number of lines of clinical investigation evidences supporting the role of western diet as a main determinant of pseudo-epidemic cause for hyperinsulinemia, which in its turn is claimed as a main determinant for the onset and prevalence of the condition known as Metabolic Syndrome characterized by the concomitant presence of insulin resistance, hypertension, and cardiovascular diseases and predisposing to type II diabetes and certain forms of cancer. Overall the article is well articulated on the clinical sections supporting the author perspective.
A major point which this reviewer finds too reductive and potentially misleading towards justifying the lines of evidence he summarized relates to the actual role of the molecular members of the insulin/IGF family of ligands, signal transducing receptors and binding proteins as he oversimplifies in chapters 2 and 3. This, in the reviewer expert opinion, needs mandatory correction in an appropriate manner which does not diminishes the value of the clinical implications of the following chapters and overall work.
Specifically, the author tends to generalize a few key established concepts of the IGF system at the molecular and functional level which are critical towards substantiating his personal suggestions at the clinical and policy levels. Namely:
On line 52: (the insulin-IGF system is formed by…) four cell-membrane receptors (insulin receptor-A, insulin receptor-B, insulin-like growth factor-I receptor (IGF-IR) and insulin-like growth factor receptor-II (IGF-II-R).….
The statement is essentially misleading in which the use of “receptor” for the IGF-II trans-membrane scavenging protein formerly referred as IGF-II receptor leads to the assumption that this molecule mediates IGF-II-activated intracellular signals comparable in any way to those mediated by the IR(A/B), the IGF1R and their hybrid forms via the presence of a well described tyrosine kinase domain which is absent in the so called IGF2R (see also Scalia et al. Cancers 2020 and Cell Cycle 2022 for an update review of the “IGF2R” function and nomenclature). In order to reflect such updated general knowledge contrary to the misleading name of IGF2R for this protein the revisited name of SPI2-6 (Scavenger Protein for IGF2 and mannose-6-phosphate) has also been suggested by some authors (Cell Cycle 2022, DOI: 10.1080/15384101.2022.2108117).
In order to avoid perpetrating incorrect assumptions on the role of the IGF2R protein in mediating IGF-II cellular effects (occurring via both IR-A and IGF1R at the cellular level), this reviewer asks that proper essential wording is added or that the statement is remodulated to reflect the established (in vivo and in vitro) experimental and clinical findings surrounding the IGF2R mechanism of action which support its actual functional role as an IGF-II scavenger/ high affinity binding protein regulating extracellular IGF2 bioavailability in vivo. As such it should not be listed among the IGFs receptor.
Furthermore, it has been extensively proven that in vivo IR and IGFR forms hybrid variants containing the alpha (ligand binding, extracellular) and beta (mostly intracellular tyrosine kinase-containing) subunits of each receptor. Such hybrids to current knowledge behave like IGFR receptors but their affinity to the insulin-IGF ligand allows to widen the traditional concept perpetrated by the author that insulin and IGFs serve separate purposes relegating to the Insulin system the pure metabolic effects while IGF-I (with no mention to the independent still unexploited role of IGF-II) would serve as both a growth factor with metabolic (anabolic) functions typical of insulin (as over-simplified in chapter 3). Therefore, this reviewer believe that the fourth receptor mentioned in line 52 should refer to the well described IR-IGF1R hybrid variant rather than to the misleading “IGF2R” protein cited.
Upon the above remarks, a suggested potential remodulation of the text in lines 52-53 and overall period reflecting extensively validated scientific findings could be made similar to:
“ The insulin-IGF system is formed by insulin, two insulin-like growth factors (IGF-I 51 and IGF-II), four cell-membrane signal transducing receptors types (insulin receptor-A (IR-A), insulin receptor-B (IR-B), insulin-like growth factor-I receptor (IGF-IR), with their respective IR-IGF1R hybrid variants) and insulin-like growth factor receptor-II (IGF-II-R) and, six IGF-binding proteins (IGFBP-1-6) including the IGF-II scavenger transmembrane protein previously known as IGF2 “receptor” (IGF2R) (reviewed in Cell Cycle 2022 DOI: 10.1080/15384101.2022.2108117 or selected refs cited herein), several IGFBP- related proteins and IGFBP proteases.”
On chapter 3, lines 70-71, in order to support the reciprocal effects and regulatory feedback of insulin and IGF1 through GH, the author cites that: “Binding of IGF-I to the IGF-I receptor primarily results in stimulation of proliferation and differentiation of cells [12]. Nevertheless, IGF-I also stimulates insulin-like actions (glucose and amino acid uptake, mRNA, and protein synthesis) in cells and enhances insulin sensitivity”. In first place in lane 75 the author writes that “Liver and fat cells do not express IGF-I receptors”. This is incorrect in that, although adult liver cells have negligible traces of IGF1R, indeed fat cells do express IGF1R in amounts that tends to decrease from pre-adipocyte to mature adipocyte differentiation (see Boucher at al PMID: 27207537) but not to the point of being quantitatively and functionally compared to what observed in liver cells. Therefore, without taking out or diminishing the author point, this statement should be rephrased.
As for the remaining part of the text and author’s perspectives and individual chapters conclusions, based on the reviewed studies along with the relevance of the message, this reviewer finds the article of interest.
Therefore, upon the mandatory revisions required on chapters 2 and 3 (which this reviewer considers major and not dispensable given their impact on current knowledge of the Insulin/IGF system and further directions to fully understands pathologies like the cited Metabolic syndrome, Diabetes and Cancer) the article could be considered for publication.

Author Response
We appreciate your comments, suggestions, and careful review of the manuscript.
We have completed our revisions based on the Reviewer ‘s helpful comments, and below we indicate point by point the changes we made to the manuscript.
Answers to comments of Reviewer 2:
1] A major point which this reviewer finds too reductive and potentially misleading towards justifying the lines of evidence he summarized relates to the actual role of the molecular members of the insulin/IGF family of ligands, signal transducing receptors and binding proteins as he oversimplifies in chapters 2 and 3. This, in the reviewer expert opinion, needs mandatory correction in an appropriate manner which does not diminishes the value of the clinical implications of the following chapters and overall work.
Specifically, the author tends to generalize a few key established concepts of the IGF system at the molecular and functional level which are critical towards substantiating his personal suggestions at the clinical and policy levels. Namely:
On line 52: (the insulin-IGF system is formed by…) four cell-membrane receptors (insulin receptor-A, insulin receptor-B, insulin-like growth factor-I receptor (IGF-IR) and insulin-like growth factor receptor-II (IGF-II-R).….
The statement is essentially misleading in which the use of “receptor” for the IGF-II trans-membrane scavenging protein formerly referred as IGF-II receptor leads to the assumption that this molecule mediates IGF-II-activated intracellular signals comparable in any way to those mediated by the IR(A/B), the IGF1R and their hybrid forms via the presence of a well described tyrosine kinase domain which is absent in the so called IGF2R (see also Scalia et al. Cancers 2020 and Cell Cycle 2022 for an update review of the “IGF2R” function and nomenclature). In order to reflect such updated general knowledge contrary to the misleading name of IGF2R for this protein the revisited name of SPI2-6 (Scavenger Protein for IGF2 and mannose-6-phosphate) has also been suggested by some authors (Cell Cycle 2022, DOI: 10.1080/15384101.2022.2108117).
In order to avoid perpetrating incorrect assumptions on the role of the IGF2R protein in mediating IGF-II cellular effects (occurring via both IR-A and IGF1R at the cellular level), this reviewer asks that proper essential wording is added or that the statement is remodulated to reflect the established (in vivo and in vitro) experimental and clinical findings surrounding the IGF2R mechanism of action which support its actual functional role as an IGF-II scavenger/ high affinity binding protein regulating extracellular IGF2 bioavailability in vivo. As such it should not be listed among the IGFs receptor.
Furthermore, it has been extensively proven that in vivo IR and IGFR forms hybrid variants containing the alpha (ligand binding, extracellular) and beta (mostly intracellular tyrosine kinase-containing) subunits of each receptor. Such hybrids to current knowledge behave like IGFR receptors but their affinity to the insulin-IGF ligand allows to widen the traditional concept perpetrated by the author that insulin and IGFs serve separate purposes relegating to the Insulin system the pure metabolic effects while IGF-I (with no mention to the independent still unexploited role of IGF-II) would serve as both a growth factor with metabolic (anabolic) functions typical of insulin (as over-simplified in chapter 3). Therefore, this reviewer believe that the fourth receptor mentioned in line 52 should refer to the well described IR-IGF1R hybrid variant rather than to the misleading “IGF2R” protein cited.
Upon the above remarks, a suggested potential remodulation of the text in lines 52-53 and overall period reflecting extensively validated scientific findings could be made similar to:
“ The insulin-IGF system is formed by insulin, two insulin-like growth factors (IGF-I 51 and IGF-II), four cell-membrane signal transducing receptors types (insulin receptor-A (IR-A), insulin receptor-B (IR-B), insulin-like growth factor-I receptor (IGF-IR), with their respective IR-IGF1R hybrid variants) and insulin-like growth factor receptor-II (IGF-II-R) and, six IGF-binding proteins (IGFBP-1-6) including the IGF-II scavenger transmembrane protein previously known as IGF2 “receptor” (IGF2R) (reviewed in Cell Cycle 2022 DOI: 10.1080/15384101.2022.2108117 or selected refs cited herein), several IGFBP- related proteins and IGFBP proteases.
A] Thank you for making this point. In the revised manuscript chapters 2 and 3 about the actual role of the molecular members of the insulin/IGF family of ligands, signal transducing receptors and binding proteins were more extensively discussed and modified in the revised manuscript (see revised manuscript Page 2, lines 51-61). In addition, it was added that IR-IGF-IR hybrids behave like IGF-IRs but show differential IGF-I vs IGF-II stimulated ac-tivity based upon the involved IR isoforms (see revised manuscript page 2, lines 94-96).
2] On chapter 3, lines 70-71, in order to support the reciprocal effects and regulatory feedback of insulin and IGF1 through GH, the author cites that: “Binding of IGF-I to the IGF-I receptor primarily results in stimulation of proliferation and differentiation of cells [12]. Nevertheless, IGF-I also stimulates insulin-like actions (glucose and amino acid uptake, mRNA, and protein synthesis) in cells and enhances insulin sensitivity”. Can the authors provide a bit more explanation on the modulating effects of IGFBPs on IGF-I action?
A] Thank you for this suggestion. The modulating effects of IGFBPs on IGF-I and IGF-II actions (See revised manuscript page 2, lines 76-79).
3]. In first place in lane 75 the author writes that “Liver and fat cells do not express IGF-I receptors”. This is incorrect in that, although adult liver cells have negligible traces of IGF1R, indeed fat cells do express IGF1R in amounts that tends to decrease from pre-adipocyte to mature adipocyte differentiation (see Boucher at al PMID: 27207537) but not to the point of being quantitatively and functionally compared to what observed in liver cells. Therefore, without taking out or diminishing the author point, this statement should be rephrased.
A] Thank you for raising this point. This paragraph was rephrased in the revised manuscript (See revised manuscript page 2, line 97 and page 3, lines 98-99).
Round 2
Reviewer 2 Report
Proper corrections have been made reflecting the state of the art on the field knowledge strengthening the authors considerations and supporting the article conclusions.